# Meta-Learning of Prompt Generation for Lightweight Prompt Engineering on Language-Model-as-a-Service

**Hyeonmin Ha[1]**     **Jihye Lee[2]**     **Wookje Han[3]**     **Byung-Gon Chun[1,2]**

[1]FriendliAI    [2]Seoul National University    [3]Columbia University

hyeonmin.ha@friendli.ai    {mmodestaa, bgchun}@snu.ac.kr

wookje.han@columbia.edu

## Abstract

Recently, many companies have been providing the capabilities of large language models as services. These Language-Model-as-a-Service (LMaaS) offerings support a variety of user tasks through in-context learning from prompts, which include instructions and demonstrations of the task. However, for users, manually crafting prompts or running automatic prompt tuning methods themselves can be demanding. Despite these challenges, LMaaS providers do not offer automatic prompt engineering methods as part of their services. One of the major obstacles to deploying them on an LMaaS is the heavy computational costs associated with automatic prompt engineering methods. These methods are typically designed to iterate through tens of thousands of examples, which impose unaffordable overheads for LMaaS providers. In this paper, we introduce MetaL-Prompt, a novel lightweight automatic prompt generation method for LMaaS. MetaL-Prompt meta-trains a prompt generation model (PGM) to enable robust learning by the language model from the contexts created by the generated prompts (i.e., in-context learning). Thanks to our meta-learning approach, a PGM can generate prompts for unseen tasks without requiring additional training for those specific tasks. Furthermore, the PGM can generate prompts with a single forward pass, significantly reducing computational costs compared to previous methods. We evaluate MetaL-Prompt on a range of unseen tasks and find that it improves performance by up to 19.4% in terms of mean F1 score on QA datasets compared to the state-of-the-art baseline P-tuning, with limited computational cost.

## 1 Introduction

Many companies have recently succeeded to train large-scale language models (LMs), and have provided these models' capabilities as services (OpenAI, 2020; Cohere, 2021; AI21Labs, 2021; Anthropic, 2023). To support diverse user tasks, such

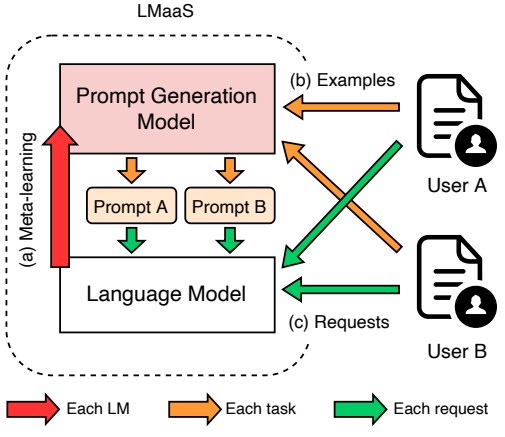

Figure 1: A workflow of MetaL-Prompt on LMaaS.

Language-Model-as-a-Services (LMaaSs) leverage in-context learning, where a model learns the task from a prompt (i.e., demonstration examples or a natural language instruction) provided by the user. While in-context learning has shown remarkable performances and flexibility on various natural language processing (NLP) tasks, it still suffers from unstable performances (Lu et al., 2022; Perez et al., 2021; Zhao et al., 2021), which motivates research on automatic prompting methods.

One of the major automatic prompt tuning methods is gradient-based prompt tuning (Liu et al., 2021; Lester et al., 2021; Li and Liang, 2021), which updates a parameterized prompt with gradients. On the other hand, gradient-free prompt tuning tackles the problem without leveraging gradients. Some gradient-free methods generate prompts with pre-trained LMs (Honovich et al., 2022; Zhou et al., 2022) or task-specific reinforcement learning (RL) agents (Deng et al., 2022), while others tune prompts using evolutionary search (Sun et al., 2022b,a). Even though such methods have shown outstanding performances, these methods typically require too heavy computation costs to be deployed on LMaaSs.

In this paper, we present MetaL-Prompt, a novel and lightweight prompt engineering method that leverages meta-learning for prompt generation. MetaL-Prompt effectively trains a prompt generation model (PGM), which is based on an LM, to facilitate robust learning of an LM from the contexts (referred to as in-context learning) created by the generated prompts (Section 3.1). The key advantage of our meta-learning approach is that it enables the PGM to generate prompts for an extensive range of unseen tasks without requiring additional task-specific training of the model. We also propose *trainable padding* (Section 3.2) to further enhance prompt generation efficiency by reducing the multi-pass generation process to a single forward pass. As a result, MetaL-Prompt significantly reduces the computational cost compared to previous methods that typically involve thousands of forward or backward passes. Prompts generated by the PGM also can be additionally tuned with gradient-based prompt tuning for further improved prompt qualities if such computation costs are allowed.

In order to utilize MetaL-Prompt for an LMaaS, the model provider initiates the process by training a PGM using our meta-learning algorithm in the offline stage (Figure 1 (a)). Once the service is operational, the trained PGM is employed to generate appropriate prompts from few-shot examples provided by a user, without the need for additional training (Figure 1 (b)). The generated prompt is then saved and consistently reused to fulfill any subsequent user requests for the same task (Figure 1 (c)).

In addition, we delve into the generation of *continuous prompts* (Section 3.3), which are real-valued vectors similar to word embeddings, in addition to the conventional discrete prompts expressed in natural language. Despite the proven effectiveness of continuous prompts in gradient-based prompt tuning (Lester et al., 2021; Liu et al., 2021; Li and Liang, 2021; Sun et al., 2022a), prior prompt generation methods (Honovich et al., 2022; Zhou et al., 2022; Deng et al., 2022) have not yet explored their potential. We compare three different approaches to continuous prompt generation and one approach to discrete prompt generation using PGMs. Through empirical analysis, we demonstrate that continuous prompts yield superior performance compared to discrete prompts, thanks to their enhanced expressiveness.

To the best of our knowledge, MetaL-Prompt stands as the pioneering approach that employs meta-learning for prompt generation utilizing few-shot examples. While FLIPPED (Ye et al., 2023) also employs meta-training for prompt generation, it is tailored for a new inference method that selects the label most likely to generate a pre-defined task instruction. FLIPPED's model scores alignment between a single data instance and a pre-defined task instruction, and it is not explicitly designed to generate prompts from few-shot examples. Other related approaches such as P-tuning v2 (Liu et al., 2022), MPT (Wang et al., 2023), and MetaPrompting (Hou et al., 2022b) also leverage meta-learning or multi-task learning for effective prompt engineering. However, their focus lies in learning improved initialization for gradient-based prompt tuning rather than the direct generation of prompts.

We evaluate MetaL-Prompt in diverse meta-learning settings where there is no overlap between training datasets and test datasets (i.e., unseen tasks). On QA datasets, MetaL-Prompt shows up to 19.4% gain in mean F1 score compared to the state-of-the-art prompt engineering methods, P-tuning, when only generated prompts without demonstrations are given for in-context learning.

## 2   Background and Related Work

**Language Model as a Service**    As large-scale language models (LMs) show astonishing performances across different tasks, several companies train such models and provide them as services (OpenAI, 2020; Cohere, 2021; AI21Labs, 2021; Anthropic, 2023) — which is often called Language-Model-as-a-Service (LMaaS) (Sun et al., 2022b). On an LMaaS, a user sends an input text to the service via the API, then she can obtain corresponding outputs from the LM.

One of the most popular methods to adapt the LM to a user's task on LMaaS is in-context learning, where a prompt — a (natural language) task instruction or/and demonstration examples for the task — is given to help the LM understand the task. Manual prompt tuning was initially studied to support in-context learning. However, manually crafted prompts require extensive human efforts to devise and sometimes show unstable and unaccountable behavior.

**Automatic Prompt Tuning**    To resolve the problems, recent studies suggest techniques to automatically search for optimal prompts. One of the

| Method | # of examples |
|---|---|
| P-tuning (Liu et al., 2021) | 56,000 |
| SoftPrompt (Lester et al., 2021) | 960,000 |
| RLPrompt (Deng et al., 2022) | 96,000 – 192,000 |
| TEMPERA (Zhang et al., 2022) | 65,536 |
| MetaPrompting (Hou et al., 2022b) | 240 |
| APE (Zhou et al., 2022) | 12,800 |
| BBTv2 (Sun et al., 2022a) | 256,000 |
| Clip-Tuning (Chai et al., 2022) | 16,000 – 32,000 |
| BDPL (Diao et al., 2023) | 32,000 – 128,000 |
| Prefix-tuning (Li and Liang, 2021) | 626,590 |
| MetaL-Prompt (ours) | 16 |

Table 1: Number of examples that each prompt tuning method is required to process for each task in the experiments of the original papers.

widely used approaches is a gradient-based method, which keeps updating prompt tokens or parameterized prompt embeddings leveraging gradients (Liu et al., 2021, 2022; Lester et al., 2021; Li and Liang, 2021; Shin et al., 2020; Hou et al., 2022b). P-tuning v2 (Liu et al., 2022), MPT (Wang et al., 2023), and MetaPrompting (Hou et al., 2022b) proposed methods to search for better initialization of the gradient-based prompt tuning using multi-task learning or meta-learning. The other line of work is a gradient-free method, which generates prompts using pre-trained LMs (Hou et al., 2022a; Honovich et al., 2022; Zhou et al., 2022), trains RL agents to generate prompts (Deng et al., 2022; Zhang et al., 2022), uses search-based optimization (Prasad et al., 2022; Sun et al., 2022b,a; Chai et al., 2022) or uses a gradient estimator (Diao et al., 2023).

**Automatic Prompt Tuning on LMaaSs** Even though automatic prompt tuning methods have proven to be effective, their adoption in Language-Model-as-a-Services (LMaaS) is currently lacking. While users can attempt to create prompts by running gradient-free prompt tuning methods using the provided LMaaS APIs, this approach poses challenges, particularly for non-experts who may find it difficult to deploy and execute such methods in their private environments.

The primary obstacle to integrating both gradient-based and gradient-free prompt tuning methods into LMaaS lies in their substantial computational costs. In Table 1, we present the number of examples (often tens of thousands) that an LM needs to process in order to optimize a prompt for each task using various prompt engineering methods. In cases where the original papers present

few-shot settings, we report the costs given 16 examples. Otherwise, we provide the actual costs as demonstrated in the papers' experiments. Considering the vast number of users that an LMaaS serves, each with unique tasks, processing such a large number of examples for a single task becomes excessively burdensome. This highlights the need for a lightweight prompt tuning method specifically tailored for LMaaS.

Note that MetaPrompting and MetaL-Prompt (ours) require meta-learning, which is a preliminary process for prompt engineering and is conducted once and not for each task. Since the meta-learning processes introduce constant costs independent of the number of users, we do not count the meta-learning processes as the costs in Table 1 which demonstrate the costs and the scalability of the baselines with respect to the number of users.

## 3 MetaL-Prompt

To address the challenges associated with automatic prompt tuning in LMaaS, we introduce MetaL-Prompt, a meta-learning approach for a lightweight prompt generation. In this approach, we meta-train a prompt generation model (PGM) (Section 3.1), which is initialized with a target language model (LM), with the objective of generating prompts that enhance the LM's contextual learning capabilities. We refer to this training process as meta-learning because the PGM learns generation of prompts that effectively induce meaningful contexts for the target LM to learn from (i.e., learning-to-learn). We also propose the use of trainable padding (Section 3.2) to alleviate the overhead of the prompt generation process, which requires multiple forward passes, during the meta-learning. Additionally, we explore various types of prompts that PGMs can generate, as discussed in Section 3.3, which have not been explored in previous prompt generation methods.

The overall workflow of an LMaaS with MetaL-Prompt is depicted in Figure 1. Initially, the model provider trains a PGM using MetaL-Prompt (Figure 1 (a)). Importantly, this training process does not impact the ongoing service as the training occurs prior to the service initiation. During the service, when a user provides a set of few-shot examples for the user's specific task, the prompt generation model generates a prompt based on these few-shot examples (Figure 1 (b)). This generated prompt is then saved and associated with the user's

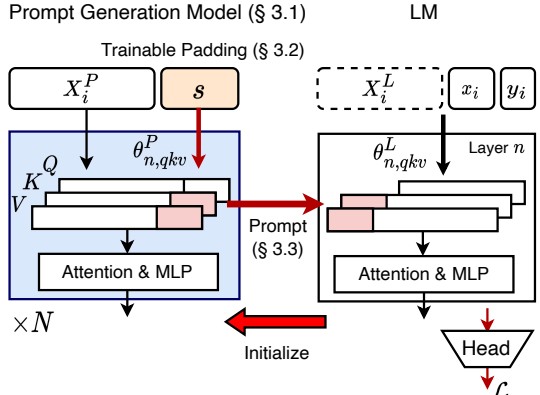

Figure 2: An illustration of meta-learning of a prompt generation model.

future requests. Finally, when the user submits a request pertaining to the task, the prompt is composed to the query, and the composed input is fed into the LM to generate the response (Figure 1 (c)).

MetaL-Prompt offers significant advancements over prior works in two key aspects: prompt quality and computation cost for prompt generation. The prompts that MetaL-Prompt generate empirically demonstrate more accurate or comparable prediction quality compared to prompts crafted by previous gradient-based or gradient-free approaches (Liu et al., 2021; Lester et al., 2021; Sun et al., 2022a; Deng et al., 2022), given limited computation budgets. In terms of computational efficiency, MetaL-Prompt surpasses previous methods by requiring only a single forward pass for prompt generation, as explained in Section 3.2. This streamlined approach is highly productive, especially when contrasted with prior approaches that necessitate an extensive number of forward or backward passes to tune or generate a prompt for a single task, as discussed in section 2. The reduction in computation cost achieved by MetaL-Prompt significantly enhances its practicality and efficiency with LMaaS.

## 3.1 Prompt generation model (PGM)

MetaL-Prompt employs meta-training to train a prompt generation model (PGM), enabling the creation of prompts that enhance in-context learning of the target language model (LM) across diverse tasks. The objective function utilized by MetaL-Prompt, as depicted in Figure 2, can be expressed

as follows:

$$\boldsymbol{\theta}_P^* = \arg\max_{\boldsymbol{\theta}_P} \sum_i p(y_i | f_{\boldsymbol{\theta}_P}(X_i^P), X_i^L, x_i; \boldsymbol{\theta}_L)$$

$$(1)$$

where $\boldsymbol{\theta}_L$ and $\boldsymbol{\theta}_P$ are the parameters of the LM and the PGM respectively, $y_i$ is the expected answer for the input text $x_i$, and $X_i^P = \{x_{i,0}^P, y_{i,0}^P, x_{i,1}^P, y_{i,1}^P, ...\}$ and $X_i^L = \{x_{i,0}^L, y_{i,0}^L, x_{i,1}^L, y_{i,1}^L, ...\}$ are concatenations of examples. $X^P$ and $X^L$ consist of examples from various NLP tasks. The prompt generation process $f_{\boldsymbol{\theta}_P}$ can be realized in different ways, and one straightforward example is choosing the most probable next tokens using probabilities predicted by the PGM. Further details regarding the prompt generation process will be discussed in-depth in Section 3.3.

The parameter $\boldsymbol{\theta}_P$ is initialized with a target LM to leverage its existing understanding of various NLP tasks. In essence, MetaL-Prompt employs Equation 1 to adapt the LM and obtain a PGM. For this adaptation process, we utilize LoRA (Hu et al., 2022), a parameter-efficient fine-tuning method, instead of full-parameter fine-tuning. Note that the original LM is frozen and only PGM is tuned during the meta-learning.

During the service phase following meta-learning, when a user provides a set of few-shot examples, MetaL-Prompt divides them into two subsets: $X^P$ and $X^L$. The PGM utilizes $X^P$ to generate an appropriate prompt. This generated prompt is then combined with the additional demonstration examples $X^L$, and the composed input is used to process future user requests. It is important to note that $X^L$ can be an empty set. In such cases, only the prompt and the input from the user's request are fed to the LM. This configuration enables the fastest inference speed due to the shorter sequence lengths.

A prompt generated by the PGM can be directly utilized for LM prompting. However, for cases where additional computational cost is acceptable, we can enhance the prompt by applying gradient-based prompt tuning methods (Lester et al., 2021; Liu et al., 2021; Li and Liang, 2021; Shin et al., 2020). In this scenario, the generated prompt serves as an initialization of gradient-based prompt tuning. We observe that this additional tuning process can further improve the performance of the prompt.

## 3.2 Trainable padding

Generative language models, such as GPT-3 (Brown et al., 2020), typically predict one token at a time within a given context, necessitating $n$ forward passes to generate $n$ tokens. This multi-pass generation process can not only introduce inefficiencies in existing serving systems (Yu et al., 2022) but also cause extra overheads to train PGMs with the objective outlined in Equation 1, which also includes generation processes.

We tackle this challenge by proposing trainable padding, which is inspired by special tokens of recent LMs and gradient-based prompt tuning (Lester et al., 2021; Liu et al., 2021). As depicted in Figure 2, MetaL-Prompt appends trainable embeddings to the given examples $X^p$, which are then fed to the PGM as part of the input. This enables the PGM to generate multiple prompt tokens simultaneously by leveraging the hidden states corresponding to each padding position. Additionally, we reparameterize the trainable padding similar to other prompt tuning methods (Liu et al., 2021; Li and Liang, 2021; Liu et al., 2022; Hou et al., 2022b), specifically employing LSTMs following the methodology of P-tuning (Liu et al., 2021).

## 3.3 Prompt design

In this section, we discuss four prompt designs — *Discrete*, *Weighted Sum*, *Hidden State*, and *Prefix* — and their generation using a PGM. While existing prompt generation methods have primarily focused on discrete prompts in natural language, we extend our investigation to include real-valued prompts (i.e., continuous prompts) as previous prompt tuning methods (Liu et al., 2021; Li and Liang, 2021; Sun et al., 2022a) have demonstrated their effectiveness.

**Discrete** is a prompt that consists of the most probable natural language tokens predicted by a PGM. To train the PGM for Discrete, we adopt Gumbel-Softmax reparameterization with discretization ("Straight-through" trick).

**Weighted Sum** is a continuous prompt obtained by multiplying the token probability predicted by a PGM with the word embeddings of a target LM. It represents the probability-weighted sum of the word embeddings.

**Hidden State** directly uses the input hidden states of the head layer in a PGM as a continuous prompt. As word embedding layers and head layers typically share the same parameters in recent

LMs, the input hidden states from the head layer have the same representations as the word embeddings. This makes them valid inputs (i.e., prompts) for the LM.

**Prefix** adds a prompt before the keys and values of transformers (Li and Liang, 2021), rather than prepends it to the inputs like the others. As depicted in Figure 2, we extract the keys and values of self-attention layers from each layer of a PGM at the position of the trainable padding, and prepend them to the keys and values of a target LM in the corresponding layers. Prefix demonstrates the best performance among all (Section 5.5), and therefore, we utilize Prefix in the following experiments (section 5).

## 4 Experimental setup

In this section, we present the setup of our experiments, which includes datasets, training and evaluation details, baselines, and models.

## 4.1 Dataset

We conduct experiments to evaluate the performance of MetaL-Prompt using the combination of CrossFIT (Ye et al., 2021) and UNIFIED QA (Khashabi et al., 2020), which consists of 142 diverse datasets. Specifically, we adopt three different task settings from MetaICL (Min et al., 2022), $cls \rightarrow cls$, $HR \rightarrow LR$, and $QA \rightarrow QA$. Each setting defines two disjoint sets — meta-learning datasets and evaluation datasets — and brief explanations of the settings are as follows.

**Classification to classification ($cls \rightarrow cls$):** Both the meta-learning datasets and the evaluation datasets encompass classification tasks.

**QA to QA ($QA \rightarrow QA$):** Both the meta-learning datasets and the evaluation datasets consist of Question-Answering (QA) tasks.

**High Resource $\rightarrow$ Low Resource ($HR \rightarrow LR$):** In this particular setting, each meta-learning dataset comprises over 10,000 training examples, while each evaluation dataset consists of fewer than 10,000 examples.

## 4.2 Evaluation

We use Macro-F1, which is a better measure than accuracy on imbalanced datasets, as our evaluation metric and report the mean scores obtained from four distinct runs. For each run, MetaL-Prompt and the baselines are given a different set of 16 examples sampled from the training split of the

evaluation dataset for prompt generation or tuning. MetaL-Prompt trains only a single PGM for all of the runs, but the PGM generates unique prompts for each run by leveraging distinct example sets. Furthermore, we explore the impact of varying example sizes and provide the corresponding results. Here, $n_P$ denotes the number of examples used for prompt generation, and $n_L$ represents the number of examples utilized for in-context learning demonstrations in addition to the tuned or generated prompts.

### 4.3 Training

MetaL-Prompt trains a PGM on the meta-learning datasets of each setting. Subsequently, for evaluation, the trained PGM is employed to generate prompts from $n_P$ examples of the unseen evaluation datasets. As we mentioned in Section 3.1, the generated prompts can be further tuned with SGD as the previous gradient-based tuning methods. If not specified, we do not adopt the additional tuning.

As described in Section 3.1, MetaL-Prompt concatenates examples to form $X_i^P$ for prompt generation and $X_i^L$ for inference, respectively. If not mentioned, we use the same $n_P$ and $n_L$ with the evaluation settings to construct $X_i^P$ and $X_i^L$ for alignment between the training and evaluation settings. More details for the training of MetaL-Prompt and the baselines are described in Appendix A and Appendix B.

### 4.4 Baselines

We compare MetaL-Prompt against five different prompt tuning baselines — P-tuning (Liu et al., 2021), SoftPrompt (Lester et al., 2021), Prefix-tuning (Li and Liang, 2021), RLPrompt (Deng et al., 2022), and BBTv2 (Sun et al., 2022a). To assess the effectiveness of the baselines on LMaaS, we evaluate their performance under constrained cost settings, roughly 10 epochs of forward and backward passes. When MetaL-Prompt adopts the further tuning of the generated prompts, we run 9 epochs of the tuning to meet the computation budget, considering the prompt generation costs. More details regarding the computation costs can be found in Appendix B.2.

### 4.5 Models

To evaluate MetaL-Prompt and the baselines, we employ autoregressive LMs: GPT2-Large (762M parameters), GPT2-XL (1.5B parameters) (Radford et al., 2019), and Llama2 (7B parameters). The motivation behind the model choice is that autoregressive models are widely utilized for LMaaS, such as OpenAI API. However, MetaL-Prompt is not limited to such models. It can support other LMs such as sequence-to-sequence models as well.

## 5 Experimental Results

We evaluate MetaL-Prompt on the setup presented in section 4. We first show the performance of MetaL-Prompt when the generated or tuned prompt is solely provided without additional demonstrations (Section 5.1). We also provide experimental computation costs to prove MetaL-Prompt's cost-efficiency in (Section 5.2). Then, we explore a trade-off between the number of examples used for in-context learning demonstrations for LMs and those for prompt generation (Section 5.3). We additionally validate whether the PGMs are capable of generalizing to inference settings that involve a different number of demonstrations from training settings (Section 5.4). Finally, we compare the performances of prompt designs we presented in Section 3.3 (Section 5.5).

### 5.1 Prompt-only In-context Learning

We present the performance of the prompts generated by MetaL-Prompt without extra demonstrations for in-context learning. In this setting, all 16 examples from a task are used for prompt generation or tuning. The inputs are composed of only the prompt and test inputs without any additional demonstrations ($n_P = 16, n_L = 0$). This setting is the most practical setting because it keeps the shortest input length. Short sequence length leads to low latency, small hidden state cache (i.e., key and value cache of transformer-based models for demonstrations or prompts), or low monetary cost for users.

As demonstrated in Table 2, MetaL-Prompt exhibits superior performance compared to the baselines, except Prefix-tuning. It achieves notable improvements of up to 19.4% on QA tasks, surpassing the state-of-the-art method (i.e., P-tuning), even with significantly lower computational costs requiring only a single forward pass. Interestingly, Prefix-tuning showcases exceptional efficacy in the limited budget setting compared to other baselines. However, MetaL-Prompt outperforms Prefix-tuning on QA tasks, and maintains comparable

| Model | Setting | BBT2 | RLPrompt | SoftP | P-tuning | Prefix | MetaL | MetaL (+Tune) |
|---|---|---|---|---|---|---|---|---|
| Large | cls→cls | 22.43 | 23.13 | 28.53 | 34.01 | 39.71 | 35.62 | **44.63** |
|  | HR→LR | 24.90 | - | 26.62 | 30.56 | 34.33 | 30.12 | **35.46** |
|  | QA→QA | 24.92 | - | 20.66 | 27.14 | 26.52 | 28.94 | **30.61** |
|  | Avg. | 24.08 | - | 25.27 | 30.57 | 33.52 | 31.56 | **36.90** |
| XL | cls→cls | 24.92 | 23.02 | 27.24 | 30.25 | 36.76 | 36.85 | **46.00** |
|  | HR→LR | 25.31 | - | 24.33 | 29.36 | 34.15 | 30.05 | **34.75** |
|  | QA→QA | 25.28 | - | 20.74 | 26.64 | 25.70 | 31.81 | **34.03** |
|  | Avg. | 25.17 | - | 24.10 | 28.75 | 32.20 | 32.90 | **38.26** |

Table 2: Comparison of evaluation results on the prompt-only setting between MetaL-Prompt and the baselines. All examples are used for prompt generation or tuning, and no additional demonstration is provided for in-context learning.

| Model | Setting | Prefix | MetaL |
|---|---|---|---|
| Llama-2 7B | cls→cls | 34.03 | **47.29** |

Table 3: Comparison of evaluation results on the prompt-only setting and Llama-2 (7B) between MetaL-Prompt and Prefix-tuning.

| Method | Computation costs | |
|---|---|---|
|  | Our constrained costs (Appendix B.2) (ms) | Original (Table 1) (hrs) |
| Prefix | 1517 | 26.4 |
| SoftP | 1880 | 50.1 |
| P-tuning | 1987 | 3.1 |
| BBT2 | 3916 | 7.0 |
| RLPrompt | 19042 | 4.2 |
| MetaL | **76** | **2.11e-5 (76 ms)** |

Table 4: Experimental computation costs of MetaL-Prompt and our baselines with our constrained computation budget and the original budgets.

performance on GPT2-XL, the larger model, and classification tasks. Notably, MetaL-Prompt benefits from its advantageously low computation costs while still delivering competitive results. In Table 3, We additionally compare the performances of Prefix-tuning and MetaL-Prompt on a larger model, Llama-2 (7B parameters), using the $cls \rightarrow cls$ setting. MetaL-Prompt outperforms our leading baseline, Prefix-tuning, on the Llama-2.

Furthermore, as discussed in Section 3.1, we can enable MetaL-Prompt to utilize comparable computation costs to the baselines (Section 4.4) by incorporating additional tuning of the generated prompts. This variant, referred to as *MetaL(+Tune)*, is indicated in Table 2. When considering the fair computation costs, MetaL-Prompt consistently achieves superior performance across various task settings and models.

Lastly, it is important to highlight that the gradient-based prompt tuning methods — Soft-Prompt, P-tuning, and Prefix-tuning — demonstrate diminishing performance as models increase in size particularly on the classification tasks. We attribute this phenomenon to the convergence speed. Due to the smaller size of the model and the prompts, the prompts tuned by these methods converge more rapidly on the smaller model, allowing them to reach better performance within the constrained computational budgets.

Conversely, MetaL-Prompt showcases its effi-

cacy on larger models, while properly benefits from additional gradient-based tuning. This means that MetaL-Prompt is expected to deliver improved performance on larger models, which is more practical, whereas other gradient-based methods may encounter limitations when applied to such models. This scalability advantage positions MetaL-Prompt as a favorable choice for scenarios involving larger models.

## 5.2 Experimental computation costs for automatic prompt engineering methods

To demonstrate the efficiency of MetaL-Prompt in computation costs, and provide experimental evidences to support computation costs of our practical setting (Section 4.4 and Appendix B.2), we measure the actual computation costs for MetaL-Prompt and our baselines. We conduct these experiments on GPT2-XL model, a single NVIDIA A100 80GB SXM, and a randomly selected dataset, WIKI-QA, with a batch size of 16.

We first measure the average time consumed for each forward and backward pass. Using the average time measured for each forward and backward pass above, we compute the actual costs required

| Method | Example Splits | |
|---|---|---|
| | (16, 0) | (12, 4) |
| | cls→cls | |
| P-tuning | 30.25 | 29.87 |
| Prefix-tuning | 36.76 | 37.07 |
| MetaL-Prompt | **36.85** | **39.33** |
| | QA→QA | |
| P-tuning | 26.64 | 26.53 |
| Prefix-tuning | 25.70 | 26.17 |
| MetaL-Prompt | **31.81** | **30.49** |

Table 5: Comparison of the performances on various distribution of examples for prompting and demonstration. We use GPT2-XL for the LM and PGMs.

| Train Setting | Test Setting | | |
|---|---|---|---|
| | (16, 0) | (12, 4) | (8, 8) |
| | cls→cls | | |
| (16, 0) | 36.85 | 31.09 | 27.75 |
| (12, 4) | 32.46 | 39.33 | 39.38 |
| | QA→QA | | |
| (16, 0) | 31.81 | 26.51 | 27.35 |
| (12, 4) | 23.56 | 30.49 | 31.31 |

Table 6: Results for transferability of the PGMs. We evaluate the PGMs for GPT2-XL on the test settings different from the training settings.

to process each task. We report two costs: our practical scenario and the settings from the original papers (Table 1).

As described in the table, the baselines consume up to several days in the original settings, and even in our practical setting, they consume several seconds. Considering these costs are measured in GPT2-XL, prompt tuning on larger models such as GPT-3 (Brown et al., 2020) requires much more costs, and the gap between MetaL-Prompt and the baselines will be larger.

### 5.3 Additional demonstrations with generated prompts

Although prompt-only is the most cost-efficient setting, model providers or users may be willing to allocate expanded computation budgets for inference or larger spatial budgets (i.e., larger key/value cache) to further increase the performance. For such situations, we explore the effect of additional demonstrations combined with the generated prompt. We keep the number of examples for each task the same but vary the ratio between $n_P$ and $n_L$. We evaluate MetaL-Prompt and two baselines, P-tuning (Liu et al., 2021) and Prefix-tuning (Li and Liang, 2021), which are the most powerful baselines in Section 5.1, on settings where $(n_P, n_L)$ is (16, 0) and (12, 4).

Table 5 presents the results of MetaL-Prompt and the baselines when additional demonstrations are provided. Notably, when given these extra demonstrations, MetaL-Prompt demonstrates further performance improvements, particularly in the cls→cls setting, with enhancements of up to 6.7%. Hence, depending on tasks, a model provider may opt to allocate a larger computation budget for inference on longer sequences or allocate additional spatial resources to cache the hidden states of the

demonstrations, thereby enhancing performance. It is worth mentioning that Prefix-tuning also benefits from these budgetary allocations in both settings, whereas P-tuning does not exhibit the same advantage.

### 5.4 Transferability to different test settings

In the previous section, we have discussed that MetaL-Prompt can further enhance performance by tailoring example splits through increased computation or spatial budgets. In this section, we further explore that a PGM trained with a specific training setting (i.e., a particular example split) is still available in different test settings. We evaluate two PGMs trained for GPT2-XL where $(n_P, n_L)$ is $(16, 0)$, $(12, 4)$. These models represent training without demonstrations and with additional demonstrations respectively. We evaluate the models on cls→cls and QA→QA with three test settings where $(n_P, n_L)$ is $(16, 0)$, $(12, 4)$, and $(8, 8)$.

As presented in Table 6, PGMs trained without demonstrations (i.e., $(n_P, n_L) = (16, 0)$) does not generalize to the other test settings. The performance decreases when examples are provided with an LM as demonstrations instead of solely utilized for prompt generation. Interestingly, PGMs trained with $(n_P, n_L) = (12, 4)$, where demonstrations are considered, show marginal or no degradation when transferred to the other test settings with demonstrations.

In summary, PGMs trained without demonstrations can not be transferred to test settings with demonstrations, and vice versa. From this observation, we notice that there exists a significant disparity between suitable prompts that are soley used without demonstrations and those that are paired with demonstrations. Exploration of this gap will be an interesting future work.

| Method | Disc | WS | HS | Prefix |
|--------|------|------|------|--------|
| F1 | 31.78 | 33.13 | 31.19 | **36.85** |

Table 7: Ablation studies on the effect of our prompt design. We evaluate various prompt designs on GPT2-XL and cls→cls.

## 5.5 Comparison between various prompt designs

In this section, we compare performances of diverse prompt designs listed in Section 3.3: *Discrete*, *Weighted Sum (WS)*, *Hidden State (HS)*, and *Prefix*. Discrete is a prompt consisting of natural language tokens, whereas Weighted Sum, Hidden State, and Prefix represent continuous prompts, which are real-valued prompts. We compare the designs on cls→cls where $(n_P, n_L)$ is $(16, 0)$.

As depicted in Table 7, Prefix exhibits the best performance among the approaches due to its ability to prepend to each layer, resulting in a larger prompt size while maintaining the same prompt length. This larger prompt size provides Prefix with a richer expressiveness compared to the other methods. Weighted Sum also demonstrates improved performance, benefiting from its enhanced expressiveness compared to the discrete prompt, which consists of natural language tokens.

However, Hidden State displays degenerated performance, even when compared to the discrete prompts. As discussed in Section 3.3, the input hidden states of the head layer have representations similar to word embeddings, but they are not identical, particularly in terms of the scale of the values. This discrepancy may cause the Hidden State prompts to deviate from the manifold of the word embeddings. We hypothesize that the degraded performance of Hidden State prompts is a result of this discrepancy. Consequently, incorporating an additional scaler to mitigate the discrepancy is expected to be beneficial in improving the performance of Hidden State prompts.

## 6 Conclusion

In this paper, we propose MetaL-Prompt, a novel and lightweight prompting method for LMaaS. MetaL-Prompt meta-trains a prompt generation model (PGM) for better in-context learning from the generated prompt, accompanying trainable padding for more efficient prompt generation. Thanks to the meta-learning, the trained PGM does not require additional training for unseen user tasks,

and it can generate a prompt with a single forward pass, allowing additional tuning of the prompt. Moreover, we explore the generation of continuous prompts using a PGM, which has not yet been discussed in the previous prompt generation studies. With the proposed designs, MetaL-Prompt achieves performance gains over five baselines up to 19.4% for mean F1 score on QA datasets with less computation cost for prompt generation than the baselines. The results support the efficiency of MetaL-Prompt in terms of model performance and computational cost.

## Limitations

**Flexibility on the number of demonstrations** As highlighted in Section 5.4, it is important to note that a prompt generation model (PGM) trained on a specific training setting (i.e., example split) cannot be directly transferred to different test settings. Consequently, when we aim to enhance prediction quality through the inclusion of additional demonstrations as on the cls→cls setting in Section 5.4, we require multiple PGMs for each specific setting, allowing for a flexible trade-off between inference speed and prediction quality by adjusting the number of demonstrations. However, training and managing multiple PGMs pose challenges for LMaaS providers.

**Prompt generation with more examples**

In our experiments, the prompt generation models (PGMs) are constrained by sequence size limits. Consequently, if a user provides an excessive number of examples, the PGMs may be unable to process such a large set if the concatenation of the examples exceeds the sequence size limit.

However, it is worth noting that recent language models have started to adopt longer sequence sizes, which helps alleviate this limitation. The incorporation of longer sequence sizes enables PGMs to handle larger sets of examples more effectively.

Additionally, we explore an iterative approach to improving prompts by concatenating a previously generated prompt with new examples. This concatenated context is then used as input to the PGM to generate an enhanced prompt, allowing the PGM to accommodate an arbitrary number of examples by iteratively processing the example subsets. This iterative approach enhances the scalability of prompt generation, empowering PGMs to process varying numbers of examples effectively.

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

| Setting | Split | |
|---|---|---|
| | Meta-learning | Eval |
| cls→cls | 43 | 20 |
| HR→LR | 61 | 26 |
| QA→QA | 37 | 22 |

Table 8: The number of datasets for each task setting. There is no overlap between the meta-learning datasets and the evaluation datasets in each setting.

## A   Training Details of MetaL-Prompt

MetaL-Prompt trains a PGM on the meta-learning datasets of each setting (i.e., cls→cls, HR→LR, and QA→QA), using up to 16,384 training examples per dataset in accordance with MetaICL (Min et al., 2022).

These settings cover a total of 133 unique tasks, which is significantly larger than the number of tasks explored in previous prompt tuning methods (Sun et al., 2022a,b; Deng et al., 2022; Prasad et al., 2022; Honovich et al., 2022; Zhou et al., 2022). The statistics of each setting are described in Table 8.

To train the prompt generation models (PGMs) of MetaL-Prompt, we utilize the AdamW optimizer (Loshchilov and Hutter, 2019) along with linear learning rate decay. The learning rate is initialized at 0.0001 for HR→LR and 0.0002 for cls→cls and QA→QA, without employing a learning rate warmup. The training epochs for cls→cls, HR→LR, and QA→QA are set to 10, 6, and 8, respectively. The prompt length for MetaL-Prompt is 20. For the additional tuning of the generated prompts, we employ an initial learning rate of 0.02 and tune them for 9 epochs to align with the computation costs associated with the gradient-based prompt tuning baselines (Section B.2). It is important to note that this additional tuning process is performed on the few-shot examples utilized by the PGM for prompt generation.

## B   Baselines

We evaluate five baselines and compare performance with MetaL-Prompt. P-tuning (Liu et al., 2021), SoftPrompt (Lester et al., 2021), and Prefix-tuning (Li and Liang, 2021) are gradient-based prompt tuning methods. RLPrompt (Deng et al., 2022) trains an RL agent to generate prompts for each task. Lastly, BBTv2 (Sun et al., 2022a) optimizes prompts using evolutionary search.

### B.1   Hyperparameters

In order to train the baselines, we use the hyperparameters listed in Table 9. For P-tuning (Liu et al., 2021), SoftPrompt (Lester et al., 2021) and Prefix-tuning (Li and Liang, 2021), we tune learning rates based on F1 scores of three classification tasks — AG News (Zhang et al., 2015b), Yelp Polarity (Zhang et al., 2015a) and TabFact (Chen et al., 2020) — that have the largest test splits among classification datasets, and we borrow other hyperparameters from the original paper. We set a prompt length of 20.

To train a policy model for RLPrompt (Deng et al., 2022), we adhere to the hyperparameters described in the original paper. In our experiments, we employ distilGPT-2 (HuggingFace, 2019; Sanh et al., 2019), as a policy model for generating optimized prompts following the original paper. To maintain consistency with the recommendations of Deng et al. (2022), we set the prompt length to 5.

Given that BBTv2 (Sun et al., 2022a) leverages the Covariance Matrix Adaptation Evolutionary Strategy (CMA-ES) (Hansen and Ostermeier, 2001; Hansen et al., 2003) during its training, we adopt all hyperparameters for CMA-ES as outlined by Sun et al. (2022a), with the exception of the population size and a budget to limit the computation cost. In line with Sun et al. (2022a), we set the prompt length to 50.

### B.2   Computation costs

In order to efficiently support Language-as-a-Service (LMaaS) with thousands of diverse user tasks, it is crucial to minimize the computation costs associated with each task. Considering our method MetaL-Prompt requires only a single forward pass with trainable padding, we adjust the number of forward and backward passes for each baseline to ensure fair comparisons, as described in Table 10.

For P-tuning, SoftPrompt and Prefix-tuning, we train our baselines for 10 epochs with a batch size of 16. It implies that, when considering one forward pass for each sequence, a cumulative total of 160 forward and backward passes are needed.

In accordance with the cost limit, RLPrompt is trained for 5 epochs. During training, RLPrompt updates its policy model using reward signals, which are logarithmic probabilities of multiple prompts, predicted by a target language model (LM). Specifically, each prompt is concatenated

| Method | P-tuning | SoftPrompt | Prefix-tuning | BBTv2 | RLPrompt |
|---|---|---|---|---|---|
| Optimizer | AdamW | AdamW | AdamW | - | Adam |
| Learning Rate | 1.6e−4 | 4e−5 | 2e−4 | - | 5e−5 |
| Learning Rate Schedule | Linear | Linear | Linear | - | Constant |
| # Epoch | 10 | 10 | 10 | 240 | 5 |
| Batch Size | | | 16 | | |
| Prompt Length | 20 | 20 | 20 | 50 | 5 |

Table 9: Hyperparameters for the baselines.

| Method | # of forward | # of backward |
|---|---|---|
| P-tuning | 160 | 160 |
| SoftPrompt | 160 | 160 |
| Prefix-tuning | 160 | 160 |
| RLPrompt | $320 \times \alpha$ | - |
| BBTv2 | 3,880 | - |

Table 10: Number of forward and backward passes for the baselines. Compared to the baselines, MetaL-Prompt requires only a single forward pass. As this number pertains to a single task, it will be linearly increased when dealing with a wide range of user tasks on LMaaS.

with every sequence and passed through the target LM to obtain rewards for updating the policy model. To calculate these rewards, we utilize 4 prompts and a batch size of 16, requiring a minimum of 64 forward passes per epoch.

Additionally, to address the limitations of RL-Prompt, which only supports single-token labels, we multiply the batch size by the number of classes to accommodate classification tasks with multi-token labels. This implies that we need a minimum of 320 forward passes, which will be further multiplied by the number of classes ($\alpha$) for each task. It is important to note that we do not consider backward passes of the policy model, as it solely updates small MLP layers (Deng et al., 2022) just before the head layer.

To assess BBTv2 in a constrained budget scenario, we designate 240 forward passes per batch for GPT2-XL and 180 forward passes for GPT2-Large, taking into account the number of hidden layers in each model (e.g., 48 layers for GPT2-XL) and a selected popsize of 5. With a batch size of 16 utilized by BBTv2, this translates to a total of 3,840 forward passes for GPT2-XL and 2,880 forward passes for GPT2-Large.