# OpenReview forum: "Meta-Learning of Prompt Generation for Lightweight Prompt Engineering on Language-Model-as-a-Service"
_EMNLP/2023/Conference — EMNLP 2023 Findings_

### Official Review · Reviewer_fHF6 · 2023-08-04

**Soundness:** 4

**Excitement:**

3: Ambivalent: It has merits (e.g., it reports state-of-the-art results, the idea is nice), but there are key weaknesses (e.g., it describes incremental work), and it can significantly benefit from another round of revision. However, I won't object to accepting it if my co-reviewers champion it.

**Paper Topic And Main Contributions:**

The paper proposes a meta-learning approach named MetaL-Prompt for prompt generation, which enables PGM to generate hints for a series of unseen tasks without additional training. MetaL-Prompt achieved the better result on QA datasets compared to the state-of-the-art baselines, even with such a small computation cost.

**Questions For The Authors:**

1. Could you provide more experimental evidences to support "computation cost for prompt generation" in Appendix B2?
2. Do you have some experimental results to argue "It can support other LMs such as sequence-to-sequence models as well.”?
3. Could you explain more about X_i^P and X_i^L mentioned in line 279-280 ? How do different settings (not only the size of sets, e.g., (16, 0) or (12, 4), but also the actual assignment, e.g., x -> X_i^P or x -> X_i^L) affect the performance of PGMs?


**Reasons To Accept:**

1. The paper is written clearly.
2. Appropriate and clear diagrams make the method understandable.
3. The experimental results and analysis are comprehensive and convincing.
4. The computation cost of the proposed method is much smaller compared to other methods, without the degradation of performance.



**Reasons To Reject:**

1. The paper didn’t mention OpenPrompt (https://github.com/thunlp/OpenPrompt), which has the same functionality as PGMs. The authors would explain why OpenPrompt can’t solve the task.
2. Case studies are missing. They could be more helpful to understand the contribution of the paper.


**Reproducibility:**

3: Could reproduce the results with some difficulty. The settings of parameters are underspecified or subjectively determined; the training/evaluation data are not widely available.

**Reviewer Confidence:**

4: Quite sure. I tried to check the important points carefully. It's unlikely, though conceivable, that I missed something that should affect my ratings.

---

> ### Author Rebuttal · Authors · 2023-08-29
>
> ## OpenPrompt
>
> OpenPrompt [1] is an integrated framework for prompt learning, which implements and collects various modules, such as training schemes, prompt templates or verbalizers. MetaL-Prompt is orthogonal to OpenPrompt and MetaL-Prompt can be a module for the framework as some of our baselines, P-tuning and Prefix-tuning.
>
> Since OpenPrompt is a collection of various modules, the availability of OpenPrompt on LMaaS depends on the modules. If an LMaaS decides to use Prefix-tuning, it has to afford the heavy computation overhead as we described in Section 2. Accordingly, OpenPrompt can employ MetaL-Prompt for its module to benefit from our light-weight prompt engineering.
>
>
> ## Case studies of MetaL-Prompt
>
> In a practical use case, we showcase the real-world applicability of MetaL-Prompt. Imagine a novelist aiming to translate their English work into another language while retaining a distinct style and tone that aligns with their unique narrative. Leveraging MetaL-Prompt within an LMaaS framework, the novelist compiles a set of specific translation examples (e.g., 16 English-Korean pairs) to define their desired style. By inputting these examples, the system generates a customized prompt that encapsulates the author's requirements. This prompt empowers the novelist to effortlessly translate numerous sentences from their work. Similarly, the author can also translate character-specific lines by generating prompts tailored to each character's persona. This seamless integration of MetaL-Prompt not only facilitates accurate translation but also facilitates the infusion of the author's creative essence into the translated narrative.
>
>
> ## Experimental computation costs for automatic prompt engineering methods.
>
> To measure experimental computation costs for Appendix B2, we first measure the average time consumed for each forward and backward pass. We conduct these experiments on GPT2-XL model, a single NVIDIA A100 80GB SXM, and a randomly selected dataset, WIKI-QA, with a batch size of 16.
>
> Using the average time measured for each forward and backward pass above, we compute the actual costs required to process each task. We report two costs on each baseline for our practical setting (Appendix B2) and the setting from the original paper (Table 1 in Section 2). For the original setting of Prefix-tuning, we use the number of training steps that the authors used for DART [2] dataset.
>
> |  | Appendix B2 (ms) |	Section 2 (hrs) |
> | ----------------|-----------------:|---------------:|
> | Prefix-tuning	| 1517 |26.40216 |
> | SoftPrompt	| 1880 |50.13719 |
> | P-tuning	| 1987 | 3.0912 |
> | BBT2	        | 3916 | 6.96164 |
> | RLPrompt	| 19042 | 4.18795 |
> | MetaL-Prompt | 76 | 2.11e-5 (76 ms) |
>
> As described in the table above, the baselines consume up to several days in the original settings, and even in our practical setting, they consume several seconds. Considering these costs are measured in GPT2-XL, prompt tuning on larger models such as GPT-3 requires much more costs, and the gap between MetaL-Prompt and the baselines will be larger.
>
> ## How different data assignments affect the performance
>
> We show the performance variance according to distribution of data into $X^P$ and $X^L$. We first randomly sample 16 examples for each eval dataset from the cls→cls setting. Then, for each run, we randomly distribute the data into $X^P$ and $X^L$, and $|X^P| = 12$ and $|X^L| = 4$. This shows $1.04$ of standard deviation in F1 score.
>
> ## Sequence-to-sequence model
>
> We will present experimental results with sequence-to-sequence models in the revision.
>
>
> [1] Ding, Ning, et al. "Openprompt: An open-source framework for prompt-learning." ACL 2022.
>
> [2] Nan, Linyong, et al. "Dart: Open-domain structured data record to text generation." NAACL 2021.

---

### Official Review · Reviewer_Y1iN · 2023-08-06

**Soundness:** 4

**Excitement:**

4: Strong: This paper deepens the understanding of some phenomenon or lowers the barriers to an existing research direction.

**Paper Topic And Main Contributions:**

This paper introduces MetaL-Prompt, a prompt generation method that addresses the challenges of the lack of an automatic prompt generation method in existing language models as a service (LMaaS) and the need for large samples for prompt tuning. The approach involves utilizing two identical Language Models (LMs), one functioning as a prompt generator and the other as a regular response generator. The weights of the LMs are frozen, and only one LoRA module is trained in the Prompt Generation Model (PGM), along with the embedding of padding as a special token. During the main experiment, the hidden states of these special tokens in PGM serve as key-value pairs to interact with the regular LM. The experimental results demonstrate that MetaL-Prompt outperforms the previous PGMs.


**Questions For The Authors:**

- Since PGM operates without direct supervision, it remains uncertain whether it can generate coherent and meaningful words. Can you offer some discrete samples of words generated by PGM?

**Reasons To Accept:**

- The method establishes a connection between the meta soft-prompt and the discrete demonstrations designed for a specific task, which is inspiring.

- The method yields comparable results without the need for fine-tuning and surpasses existing prompt generation methods when fine-tuned with considerable computing resources.

- The paper extensively investigates the utilization of hidden states acquired from special tokens, which act as prompts.

- Supplementary experiments are detailed, particularly focusing on the varied proportions of prompts used for PGM and LM during training and testing, thereby enhancing the effectiveness of PGM.


**Reasons To Reject:**

- If possible, the existing method can be easily extended to a larger model. GPT2 is slightly weak as a basic model, and the method will be more convincing under a more powerful base model.

- This method demonstrates clear advantages under low-resource conditions. However, as an in-context learning approach, when abundant samples are accessible, prefix-tuning might be a more straightforward alternative, while this method could face challenges in allocating a large number of demonstrations within a single sample.



**Reproducibility:**

4: Could mostly reproduce the results, but there may be some variation because of sample variance or minor variations in their interpretation of the protocol or method.

**Reviewer Confidence:**

3: Pretty sure, but there's a chance I missed something. Although I have a good feel for this area in general, I did not carefully check the paper's details, e.g., the math, experimental design, or novelty.

---

> ### Author Rebuttal · Authors · 2023-08-29
>
> ## MetaL-Prompt on a larger model
>
> To prove the real-world applicability of MetaL-Prompt, we additionally compare the performances (averaged F1 score) of Prefix-tuning and MetaL-Prompt on a larger model, Llama-2 (7B parameters) [1], using the cls→cls setting.
> For Llama-2, We follow the same settings with Section 5.1, Appendix A, and Appendix B2, but we adjust the learning rate to 4e-4 for MetaL-Prompt and 5e-5 for Prefix-tuning. Please note that the results of GPT-2 models match those we reported in our paper.
>
>
> |  Model | Prefix | MetaL |
> | --------- |--------:|--------:|
> | GPT2-Large (0.77B)  | 39.71 | 35.62 |
> | GPT2-XL (1.5B) |  36.76 | 36.85 |
> | Llama-2 (7B) | 34.03 | 47.29 |
>
> MetaL-Prompt outperforms our leading baseline, Prefix-tuning, on the Llama-2. Notably, this experiment supports our investigation in Section 5.1 on Prefix-tuning's diminishing performance as models increase in size. This is attributed to the inherent challenge of effectively training a continuous prompt with a larger parameter space within our practical training configuration, consequently constraining the number of feasible training steps. Conversely, MetaL-Prompt showcases its efficacy in such settings.
>
>
>
> ## High-resource setting
>
> Developers are continually introducing LMs with longer sequence capabilities, like Anthropic's Claude 2 [2], accommodating up to 100K tokens. This feature enables MetaL-Prompt to effectively leverage a substantial example pool for prompt generation. Although Prefix-tuning may still outpace MetaL-Prompt in the utilization of high data resources on these models, it's vital to consider that Prefix-tuning necessitates a separate training system, distinct from the inference service. It also incurs higher computational costs, as discussed in Section 2, restraining Prefix-tuning's scalability.
>
> Should the feasibility of such auxiliary deployment and the associated computational overhead be acknowledged, MetaL-Prompt can seamlessly incorporate additional gradient-based prompt refinement, as outlined in Sections 3.1 and 5.1. The ensuing performance enhancement becomes apparent in the "MetaL (+Tune)" results presented in Table 2.
>
>
> ## Discrete prompt examples
>
> We present concrete examples of prompts that a PGM generates. The PGM is based on GPT2-XL, and is trained to generate “Weighted Sum” prompts since “Prefix” is hard to interpret. To make the continuous prompts human-readable, each prompt embedding $p_i$ is interpreted to the token whose weight $w_{i,j}$ is the highest in the vocabulary $\mathcal{V}$, and we also report the weights. Note that the sum of the weights for each prompt embedding equals 1: $\sum_{j}{w_{i,j}} = 1 (j < |\mathcal{V}|)$.
>
> Medical Question Pairs (MQP) [3]
>
> ```
> ['itional:1.0', 'itional:1.0', 'itional:1.0', 'itional:1.0', 'itional:0.99', 'itional:0.22', ' Diversity:0.05', ' Diversity:0.06', ' Diss:0.15', ' Similar:0.78', '\xa0:0.74', ' \xa0:0.47', ' \xa0:0.95', ' \xa0:0.97', 'ib:1.0', 'ib:1.0', 'ib:1.0', 'ib:1.0', 'ib:1.0', 'ib:1.0']
> ```
>
> AG News [4]
> ```
> ['itional:1.0', 'itional:1.0', 'itional:1.0', 'itional:1.0', 'itional:1.0', 'itional:0.19', ' Sports:0.08', ' Sports:0.1', ' Sports:0.19', ' Sci:0.35', '\xa0:0.77', '\n\xa0:0.73', ' \xa0:0.93', ' \xa0:0.96', 'ib:1.0', 'ib:1.0', 'ib:1.0', 'ib:1.0', 'ib:1.0', 'ib:1.0']
> ```
>
> SuperGLUE CommitmentBank (CB) [5]
> ```
> ['itional:1.0', 'itional:1.0', 'itional:1.0', 'itional:1.0', 'itional:1.0', 'anim:0.41', 'anim:0.07', 'anim:0.06', ' hypothesis:0.05', ' contradiction:0.69', '\xa0:0.6', '\n\xa0:0.75', ' \xa0:0.9', ' \xa0:0.96', 'ib:1.0', 'ib:1.0', 'ib:1.0', 'ib:1.0', 'ib:1.0', 'ib:1.0']
> ```
>
> Remarkably, we notice a consistent mapping of the beginnings and endings of prompts to frequently occurring tokens, "itional" and "ib," both of which bear remarkably high weights. We hypothesize that these prompt embeddings potentially signal the commencement of a sequence and serve as a delimiter between the prompt and a query example. Additionally, our observations indicate a slight bias towards the verbalized classes of each dataset within the midsection of the prompts. However, the weights associated with these tokens are not overly substantial, suggesting that each embedding is a collaborative representation of multiple tokens.
>
>
> [1] Touvron, Hugo, et al. "Llama 2: Open foundation and fine-tuned chat models." 2023.
>
> [2] https://www.anthropic.com/index/claude-2
>
> [3] McCreery, Clara H., et al. "Effective transfer learning for identifying similar questions: matching user questions to COVID-19 FAQs." ACM 2020.
>
> [4] Zhang, Xiang, Junbo Zhao, and Yann LeCun. "Character-level convolutional networks for text classification." NIPS 2015.
>
> [5] De Marneffe, Marie-Catherine, Mandy Simons, and Judith Tonhauser. "The commitmentbank: Investigating projection in naturally occurring discourse." Sinn und Bedeutung. Vol. 23. No. 2. 2019.

---

### Official Review · Reviewer_SZaX · 2023-08-11

**Typos Grammar Style And Presentation Improvements:** 1. The diagrams presented are somewha…
**Soundness:** 2

**Excitement:**

3: Ambivalent: It has merits (e.g., it reports state-of-the-art results, the idea is nice), but there are key weaknesses (e.g., it describes incremental work), and it can significantly benefit from another round of revision. However, I won't object to accepting it if my co-reviewers champion it.

**Missing References:**

Self-supervised Meta-Prompt Learning with Meta-Gradient Regularization for Few-shot Generalization.

**Paper Topic And Main Contributions:**

The paper presents a novel framework called MetaL-Prompt to solve the issue of absence of prompt tuning methods on LMaaS platforms.
The key insight is the  lightweight automatic prompt generation method where an LM robustly learns from contexts induced by the created prompt, in a manner of learning to learn. Experiments on diverse unseen tasks show the superiority of the proposed model.

**Questions For The Authors:**

In line 169, "We do not count the meta-learning processes as the costs to show the scalabilities of the baselines." This sentence is confusing, can you paraphrase it?

In line 276, in the Equation 1, can you give any introduction and explanation about the $y_i$ ?

In line 338, "This enables the PGM to generate multiple prompt tokens simultaneously by leveraging the hidden states corresponding to each padding position." How does process work? Can you give more details?


**Reasons To Accept:**

The paper showcases a substantial advancement in experimental outcomes, signifying its robustness and credibility. Simultaneously, the research problem's high practical relevance underscores its potential real-world impact, making the paper's contributions valuable for both academia and practical applications.

**Reasons To Reject:**

1. The paper appears to have limited innovation and technical depth, leading to repeated restatements in various sections to compensate for this deficiency. Furthermore, the overall effort appears to be insufficient. The writing quality is moderate, and the graphical representations are relatively basic and lacking refinement, which may impede comprehension.

2. The paper's lack of a substantial baseline comparison undermines the persuasiveness of its experimental results, consequently casting doubt on the validity of the significant improvements claimed. Despite the presence of only one formula in the entire manuscript, this equation remains inadequately explained.

3. Regrettably, Sections 3.1 and 3.2, which encompass the most crucial methodology, suffer from a lack of clarity, detailed explanation, and adequate length, with a surplus of repetition and redundancy in less critical areas. Both the diagram clarity and corresponding textual descriptions are lacking, with a repeated focus on supposed advantages.

4. Furthermore, the discrepancy between the paper's reliance on smaller models for experimentation and the industry's practical utilization of larger models diminishes the real-world applicability and relevance of the paper's experimental outcomes.

5. No code and data provided. Experimental results are not reproducible.

Consequently, due to these inadequacies and inconsistencies, the paper is not deemed suitable for publication.

**Reproducibility:**

2: Would be hard pressed to reproduce the results. The contribution depends on data that are simply not available outside the author's institution or consortium; not enough details are provided.

**Reviewer Confidence:**

4: Quite sure. I tried to check the important points carefully. It's unlikely, though conceivable, that I missed something that should affect my ratings.

---

> ### Author Rebuttal · Authors · 2023-08-29
>
> ## MetaL-Prompt on a larger model
>
> To prove the real-world applicability of MetaL-Prompt, we additionally compare the performances (averaged F1 score) of Prefix-tuning and MetaL-Prompt on a larger model, Llama-2 (7B parameters) [1], using the cls→cls setting.
> For Llama-2, We follow the same settings with Section 5.1, Appendix A, and Appendix B2, but we adjust the learning rate to 4e-4 for MetaL-Prompt and 5e-5 for Prefix-tuning. Please note that the results of GPT-2 models match those we reported in our paper.
>
>
> |  Model | Prefix | MetaL |
> | --------- |--------:|--------:|
> | GPT2-Large (0.77B)  | 39.71 | 35.62 |
> | GPT2-XL (1.5B) |  36.76 | 36.85 |
> | Llama-2 (7B) | 34.03 | 47.29 |
>
> MetaL-Prompt outperforms our leading baseline, Prefix-tuning, on the Llama-2. Notably, this experiment supports our investigation in Section 5.1 on Prefix-tuning's diminishing performance as models increase in size. This is attributed to the inherent challenge of effectively training a continuous prompt with a larger parameter space within our practical training configuration, consequently constraining the number of feasible training steps. Conversely, MetaL-Prompt showcases its efficacy in such settings.
>
> ## Technical depth
>
> Our technical foundation can be attributed to our unique bi-level approach employed in the training of our prompt generation model (PGM). Our training scheme involves a synergistic combination of two core challenges: prompt generation and in-context learning. This strategy empowers us to enable fully gradient-based training for the PGM, eliminating the need for supervision on prompts. In contrast to MetaL-Prompt, RLPrompt [2] and TEMPERA [3] resort to partially gradient-based training using ad-hoc signals (i.e., rewards), which does not leverage gradients for in-context learning. FLIPPED [4] heavily relies on supervised prompt information.
>
>
> ## Baselines
>
> Our experiments encompass a diverse array of methodologies --- RL-based, search-based, and gradient-based approaches, which also covers both major categories, gradient-based and gradient-free methods.
>
> Thank you for noting the missing reference, SUPMER [5]. Unlike MetaL-Prompt, SUPMER necessitates mandatory training from a non-task-specific meta-initialized prompt, which is acquired to enhance gradient-based prompt tuning. However, this requirement places a substantial burden on deployment due to the need for a training system and the associated computation costs. In contrast, MetaL-Prompt directly generates task-specific prompts, resulting in more lightweight costs. We will compare the performances of SUPMER and MetaL-Prompt in the revision.
>
>
> ## Clarification and writing
>
> * Computation cost of MetaL-Prompt
>     * We exclude the meta-learning process cost in Table 1. This emphasizes the scalability gap regarding user numbers between previous methods and MetaL-Prompt. Our training precedes LMaaS deployment, resulting in a constant cost unaffected by user numbers.
> * $y_i$ in Equation 1
>     * $y_i$ represents the expected answer for the input text $x_i$. For example, in natural language inference, $y_i$ denotes the verbalization of entailment, neutral, or contradiction, and in QA tasks, $y_i$ is the answer for the question $x_i$.
> * Mechanism of trainable padding
>     * Decoder transformer models, as well as Encoder-Decoder transformer models, are designed to generate a single token given input text by calculating probability only on the final token position. Trainable paddings enable computing embeddings or probabilities for each padding position.
> * In the upcoming revision, we will also address duplicated explanations and presentations of the diagrams and equations, incorporating the clarifications mentioned earlier.
>
>
>
> ## Code and data release
>
> The datasets utilized in our paper align with those employed by MetaICL [6]. These datasets are publicly accessible through the MetaICL repository [7]. As stated in the paper, we have adhered to the identical task settings with MetaICL. Moreover, we are in the process of preparing the release of our code after the anonymity period.
>
>
> [1] Touvron, Hugo, et al. "Llama 2: Open foundation and fine-tuned chat models." 2023.
>
> [2] Deng, Mingkai, et al. "Rlprompt: Optimizing discrete text prompts with reinforcement learning." EMNLP 2022.
>
> [3] Zhang, Tianjun, et al. "Tempera: Test-time prompting via reinforcement learning." ICLR 2023.
>
> [4] Ye, Seonghyeon, et al. "Guess the instruction! making language models stronger zero-shot learners." ICLR 2023.
>
> [5] Pan, Kaihang, et al. "Self-supervised Meta-Prompt Learning with Meta-Gradient Regularization for Few-shot Generalization." 2023.
>
> [6] Min, Sewon, et al. "Metaicl: Learning to learn in context." NAACL 2022.
>
> [7] https://github.com/facebookresearch/MetaICL

---

### Meta-Review · Area_Chair_vm6C · 2023-09-19

**Recommendation:** 4

**Metareview:**

This paper introduces MetaL-Prompt, a prompt generation method that addresses the challenges of the lack of an automatic prompt generation method in existing language models as a service (LMaaS) and the need for large samples for prompt tuning. With this meta-learning based approach, a prompt generator model can generate prompts for a series of unseen tasks without additional training.

Pros:

Well motivated with a timely and a very relevant research topic

The paper establishes a connection between the meta soft-prompt and the discrete demonstrations designed for a specific task

Yields comparable results without the need for fine-tuning and surpasses existing prompt generation methods when fine-tuned with considerable computing resources.

The idea of utilizing hidden states acquired from special tokens as prompts is interesting

Cons:

The authors have used GPT-2 large and XL .. these are both fairly weak and outdated models and significantly smaller in size compared to SoTA models today. In order to make this work more relevant and more convincing the authors need to show that their proposed method works even for large enough LLMs

The paper would benefit from some case studies to understand the effectiveness of the method

This method demonstrates clear advantages under low-resource conditions. But it would be good to understand how it performs under the more common high resource scenario where abundant samples are available (this point also has been raised by multiple reviewers)

---

### Decision · Program_Chairs · 2023-10-07

**Decision:**

Accept-Findings

**Comment:**

This paper introduces MetaL-Prompt, a prompt generation method that addresses the challenges of the lack of an automatic prompt generation method in existing language models as a service (LMaaS) and the need for large samples for prompt tuning. With this meta-learning based approach, a prompt generator model can generate prompts for a series of unseen tasks without additional training.

Pros:

Well motivated with a timely and a very relevant research topic

The paper establishes a connection between the meta soft-prompt and the discrete demonstrations designed for a specific task

Yields comparable results without the need for fine-tuning and surpasses existing prompt generation methods when fine-tuned with considerable computing resources.

The idea of utilizing hidden states acquired from special tokens as prompts is interesting

Cons:

The authors have used GPT-2 large and XL .. these are both fairly weak and outdated models and significantly smaller in size compared to SoTA models today. In order to make this work more relevant and more convincing the authors need to show that their proposed method works even for large enough LLMs

The paper would benefit from some case studies to understand the effectiveness of the method

This method demonstrates clear advantages under low-resource conditions. But it would be good to understand how it performs under the more common high resource scenario where abundant samples are available (this point also has been raised by multiple reviewers)